# Developing a Global Community of Practice for Pharmacy Workforce Resilience—Meet GRiT

**DOI:** 10.3390/pharmacy9020110

**Published:** 2021-06-10

**Authors:** Karen Whitfield, Vibhuti Arya, Zubin Austin, Dalia Bajis, Catriona Bradley, Bronwyn Clark, Betty Exintaris, Kirsten Galbraith, Maguy Saffouh El Hajj, Kat Hall, Louise Hughes, Sue Kirsa, Catherine Langran, Efi Mantzourani, Kyle John Wilby, Sarah Willis

**Affiliations:** 1School of Pharmacy, University of Queensland, Brisbane, QLD 4072, Australia; 2College of Pharmacy and Health Sciences, St. John’s University, Queens, NY 11439, USA; aryav@stjohns.edu; 3Leslie Dan Faculty of Pharmacy and the Temerty Faculty of Medicine, University of Toronto, Toronto, ON M52 3M2, Canada; zubin.austin@utoronto.ca; 4Faculty of Medicine and Health, School of Pharmacy, University of Sydney, Sydney, NSW 2006, Australia; dalia.bajis@sydney.edu.au; 5Irish Institute of Pharmacy, D02 FP84 Dublin, Ireland; catrionabradley@iiop.ie; 6Australian Pharmacy Council, Canberra, ACT 2609, Australia; bronwyn.clark@pharmacycouncil.org.au (B.C.); sue.kirsa@monashhealth.org (S.K.); 7Faculty of Pharmacy and Pharmaceutical Sciences, Parkville Campus, Monash University, Parkville, VIC 3052, Australia; betty.exintaris@monash.edu (B.E.); kirstie.galbraith@monash.edu (K.G.); 8College of Pharmacy, Qatar University, Doha, Qatar; maguyh@qu.edu.qa; 9School of Pharmacy, University of Reading, Reading RG6 6AH, UK; k.a.hall@reading.ac.uk (K.H.); c.a.langran@reading.ac.uk (C.L.); 10Cardiff School of Pharmacy and Pharmaceutical Sciences, Cardiff University, Cardiff Cf10 3NB, UK; HughesML@cardiff.ac.uk (L.H.); MantzouraniE1@cardiff.ac.uk (E.M.); 11Pharmacy Department, Monash Health, Clayton, VIC 3168, Australia; 12College of Pharmacy, Dalhousie University, Halifax, NS B3H 4R2, Canada; kyle.wilby@dal.ca; 13Centre for Pharmacy Workforce Studies, Division of Pharmacy and Optometry, Faculty of Biology, Medicine and Health, The University of Manchester, Manchester M13 9PT, UK; sarah.willis@manchester.ac.uk

**Keywords:** resilience, pharmacy profession, GRiT, pharmacist, pharmacy workforce, health human resources, international collaboration, well-being

## Abstract

Workforce resilience in pharmacy is required to ensure the practice, education, and administrative systems remain viable and sustainable over time and when facing challenges. Whether it is addressing burnout of pharmacists or students, or the structure and policies/procedures of employment and professional organizations, working to increase resilience across all individuals and sectors is essential to relieve pressure and promote better well-being, especially during the recent pandemic. The purpose of this article is to describe the development of a community of practice global group focused on development of resilience within the pharmacy workforce that is inclusive of students, pharmacy interns/preregistration and registered pharmacists. The steering group meets monthly and has representation of 24 members across eight countries. Members meet to discuss pertinent issues they are facing in practice, as well as to share and progress ideas on education, research, and practice initiatives. To date, members have collectively implemented resilience training in pharmacy education, researched burnout and resilience in both students and pharmacists, and facilitated international collaborations both within and outside core group members. Future activities will focus on strengthening the community of practice in order to harness the power of the collective.

## 1. Introduction

There are increasing calls to investigate and develop workforce resilience across the health system as a priority, especially in light of COVID-19 [1,2,3]. Students and professionals are facing greater pressures with increasing stress levels that are known to deplete resilience and negatively impact well-being. At the same time, some employers and professional organizations have policies and procedures that may negatively impact workforce productivity by depleting resilience of both individuals and systems.

University students consistently report experiencing higher levels of mental distress compared to the general population [4]. Studies have highlighted high levels of mental distress in this population, specifically sleeping problems, depression, anxiety, and suicide risk [5]. University students who experience mental distress are likely to experience negative consequences such as impaired cognitive functioning, poor academic performance and burnout [6,7]. For health professionals, higher levels of resilience and well-being can mitigate burnout and other effects of work-related stress [8,9]. There is an increased research focus on the issue of resilience and how it influences pharmacy students and pharmacists’ success and performance. Psychometric resilience scales have been developed specifically for pharmacy students and a recent US study investigated factors influencing pharmacy students and pharmacists’ resilience and well-being [10,11].

Resilience is at the core of the WHO European policy framework for health and well- being—Health 2020—and the United Nations Sustainable Development Goals [12]. The WHO distinguishes between three levels of resilience (individual, community and system) and their implications for health. It also describes four capacities of resilience—absorptive, adaptive, anticipatory and transformative—which can be applied at all three levels. As global interest in workforce resilience continues to grow, it is important for those working in the space to share findings and work collaboratively to develop initiatives and improve intended outcomes. One potential way of doing this is through forming communities of practice. Communities of practice are groups of people who share an interest in a concept or topic and learn how to better address it by interacting regularly to discuss issues and share practice experience, while aiming to build a sense of community [13,14]. It is suggested that communities of practice develop their practice through a wide range of activities. These may include problem solving, request for information, seeking experience, reusing assets, coordination and strategy, building an argument, growing confidence, discussing developments, documenting projects, and mapping knowledge and identifying gaps [15]. Here, we describe the development of a global community of practice interested in the development of resilience within the pharmacy workforce that is inclusive of students, pharmacy interns/preregistration and registered pharmacists.

## 2. Materials and Methods

The concept of a global resilience team (GRiT) started in September 2019 following delivery of a workshop on ‘Developing Resilience in Undergraduate Pharmacy Students’ at the International 10th Biennial Monash Pharmacy Education Symposium 2019. A call for expressions of interest to contribute was offered at the end of the workshop. During the first meeting, representation, aims, purpose, governance and terms of reference were discussed. The Chair and steering committee evolved over the next 2–3 months. A global community of practice was formed, and the purpose of GRiT was defined as a forum to learn, share ideas and to provide a support network to those interested in developing a greater understanding of resilience in the profession. The group was intended to provide a platform for people to collaborate on a range of initiatives, projects and research studies in the field of resilience and to share their findings. Specific areas of interest included resilience in undergraduate students, pharmacy interns/preregistration students, postgraduate pharmacists, preceptors and educators. Pharmacy interns or preregistration pharmacists are pharmacy graduates who are completing a period of supervised practice before being eligible to apply for general registration.

### 2.1. Membership

The decision was made that 2 types of membership were to be included—steering group members and general group members. The steering group consists of 24 members working collaboratively. Eligibility for membership was agreed to be via invitation and decided by the group at the monthly meetings. The overall responsibility of the steering group was to provide support, oversight and strategic direction for GRiT. General members are interested individuals, who can be invited to join the meetings to present work or seek advice from GRiT. The decision was made to keep the steering group small initially with global representation, to enable GRiT to become established and a strategic plan formulated. Inviting general members to present has added depth to discussions. As GRiT evolves and develops, we expect changes to be made to the governance and structure of the membership.

### 2.2. Specific Activities

GRiT meets monthly via a video conferencing platform. Meetings have an agenda and minutes are prepared and distributed by the Chair or a designated steering group member in advance. Meeting are structured with inclusion of an informal presentation by a steering group or general group member of proposed initiatives or actual work being undertaken. Meetings include an opportunity for open dialogue and discussion to share ideas/knowledge and seek advice on proposed or ongoing initiatives between group members.

## 3. Results

To date, the global community of practice GRiT has met every month since September 2019. The Chair is supported by a steering group, composing 23 members from the following countries USA, Canada, UK (England and Wales), Ireland, Bahrain, Qatar, Australia and New Zealand.

Communities of practice develop through a range of activities, as such the community should be dynamic, allowing participation and learning on the part of everyone [15]. Such activities can be found in Table 1 together with overarching examples from GRiT.

Community of practice activities are described in more detail below together with several achievements of GRiT and GRiT members.

### 3.1. Incorporating Resilience Education into Undergraduate Pharmacy Programs

#### 3.1.1. Faculty of Pharmacy and Pharmaceutical Sciences, Monash University

Involvement with GRiT created the inspiration to embed the notion of resilience and the development of resilience skills into the undergraduate programs at the Faculty of Pharmacy and Pharmaceutical Sciences (FPPS), Monash University. The recently redesigned educational programs had a strong emphasis on skill development and including resilience and grit as core competencies was seen as important. In 2019/2020, two interventions were implemented and described below.

First year skills coaching

The skills coaching program at FPPS enables students to focus on their skill development. Eight core skills underpinning the pharmacy program are: problem solving, oral communication, written communication, empathy, reflective practice, integrity, teamwork, and inquiry. Students are asked to reflect on their experience and development. Skills coaches provide feedback on the reflections and from there the student identifies specific actions to work on.

Small groups of students (10–12) regularly meet with a dedicated skills coach. At each meeting, the skills coach facilitates a discussion and/or introduces an activity to the students. In 2019, resilience was included as one of the topics in the first-year skills coaching program. Students were asked to define resilience, then asked to consider some of the ways in which their resilience had been impacted, and to consider strategies to improve their resilience. In both 2019 and 2020, it was observed by facilitators that the topic of resilience generated meaningful discussion between the students and between students and skills coaches.

Fourth year students’ workshop

At the beginning of 2020, a tailor-made two-hour resilience workshop for fourth year students was designed and implemented. Prior to this class, the notion of resilience had not been presented to this cohort of students. Given that the students would be graduating soon, it was deemed necessary to start the conversation to ensure students were aware of how to build on their existing resilience prior to entering the workforce as professionals.

The workshop included a mix of individual and team-based activities centered on defining resiliency, evaluating previous and existing resiliency, and developing a resiliency plan. The workshop was supported with a resilience diary (http://www.resilienceagenda.com, accessed on 8 June 2021) to reinforce the concepts discussed over the entire year. Anecdotally, there was a lot of energy in the room and the feedback from the class was overwhelmingly positive.

#### 3.1.2. School of Pharmacy University of Otago

As a result of collaboration between GRiT members and sharing previously developed material on resilience, a 90 min workshop was developed on student balance and embedded into the 4th year undergraduate curriculum [16]. Adapted materials were transformed for the workshop and divided into two parts. The first part used the coaching tools of ‘Wheel of Life’ and ‘Anti-Goals’ with an aim to have students identify their current life priorities and the ways in which they could work to avoid resilience-depleting scenarios, for example ‘worst possible day’ in their daily lives. The second activity involved facilitated small group work in which students were asked to review cases created from adapted materials. The cases were intended to elicit their coping and self-regulation strategies when faced with situations of adversity, for example racism, negative feedback and burnout. Student responses showed traditional coaching life priorities (as identified by the Wheel of Life) may need to be modernized to account for generational changes in life priorities. Responses also indicated that student’s value being prepared for their daily activities to avoid resilience depletion. This type of activity is an example of one that could be replicated across global settings to help inform undergraduate resilience training and modernize coaching tools for younger generations.

### 3.2. Incorporating Resilience Education into Continuing Professional Development for Pharmacists

#### The Irish Institute of Pharmacy

The Irish Institute of Pharmacy (IIOP) supports and quality assures continuing professional development (CPD) activities on behalf of the Pharmacy Regulator, the Pharmaceutical Society of Ireland (PSI). Self-management and workplace-management are included in the Core Competency Framework for Irish pharmacists [17]. The IIOP has supported the development of these skills through training programs focused on management and leadership skills, and through the ongoing development and facilitation of the IIOP peer-support network. At the start of the COVID-19 pandemic, the IIOP sought direction from this peer network about how best to support pharmacists in practice. Clear themes emerged from this engagement. Pharmacists expressed concern about burnout and requested resources to assist them in increasing their resilience. They also sought virtual fora to facilitate connection with colleagues and to enable the efficient sharing of relevant, contemporaneous information relating to COVID. In response, new approaches were developed by the IIOP including the establishment of a virtual COVID hub and the development of a webinar series (available through www.iiop.ie, accessed on 8 June 2021). Webinar topics addressed a range of issues such as resilience, burnout and stress management as well as topical clinical issues. Sixteen webinars were hosted between April and December 2020, with a total of 5465 registrants (average of 342 per webinar). Recordings of the webinars were available for viewing on the IIOP website afterwards with an average of 215 views per webinar (Range 35–733). Rich insights have been gained from the webinar discussions and feedback from participants. As a result, new approaches to supporting CPD are emerging within the IIOP to meet the evolving needs of practitioners.

A needs analysis was undertaken by the IIOP’s COVID-Hub Mental Health working group in October 2020, to better understand the topic of stress management amongst pharmacists. This provided further insights into pharmacist’s experiences and perceptions of stress, and the types of interventions required to enhance resilience. The IIOP is engaging with members of the GRiT collaborative to explore new approaches to support pharmacists’ continuing professional development in this area. In addition, CPD resources for practitioners will be the focus of the first funded projects within GRiT, which will assist in raising the importance of resilience for practicing pharmacists.

### 3.3. Conducting Research in Resilience and Burnout within the Profession

#### College of Pharmacy, Qatar University, Doha, Qatar

Involvement in GRiT provided the idea and motivation to conduct research in the area of burnout and resilience among the pharmacy workforce in Qatar. Burnout is a psychological syndrome characterized by three dimensions: emotional exhaustion, depersonalization, and decreased feeling of personal accomplishment [18]. Burnout can lead to deteriorated physical and mental health which may result in devastating negative consequences related to work performance, quality of care, and patient safety [19]. The World Health Organization [12] declared the COVID-19 outbreak a public emergency of international concern. As the most accessible healthcare providers, in addition to their day-to-day activities, community pharmacists were expected to perform several responsibilities in order to fight and contain the COVID-19 outbreak [20]. The pandemic has put healthcare professionals including pharmacists at a high risk of psychological stress and exacerbation of burnout. A cross sectional survey of community pharmacists in Qatar is currently being conducted to assess their level of burnout and resilience during COVID-19 pandemic. Burnout is measured using the gold standard tool Maslach Burnout Inventory: Human Services Survey (MBI-HSS). The MBI is a validated tool used to measure all three dimensions of burnout [21]. Community pharmacists’ resilience is being measured using the Connor–Davidson Resilience Scale (CD-RISC10). The survey was opened in November 2020. So far, over 120 community pharmacists in Qatar have completed it. The survey closed for data analysis in January 2021. The study findings will help form future strategies and interventions to support community pharmacists to mitigate burnout and improve their resilience and well-being in Qatar and the wider region.

### 3.4. Exploring Resilience of Pharmacy Students and Faculty Members in the Eastern Mediterranean Region (EMR)

Using shared learning from the GRiT membership, a study to explore pharmacy student and faculty member resilience across schools and colleges of pharmacy in the Eastern Mediterranean Region (EMR) was initiated. The EMR is one of the WHO’s six regions and has a population of approximately 679 million people. Whilst the EMR shares some similarities, including a common language in parts and a comparable cultural milieu, it is a highly nuanced and dynamic region, both politically and socio economically. Up until recently, studies on resilience in higher education have found dominance in Western-based research, with limited focus on countries within the EMR [19]. Research within the EMR is especially timely given that global initiatives, aimed at developing and implementing systems for improving prevention and early interventions for mental health problems among university students, are underway.

The EMR’s resilience in pharmacy education study aims to investigate factors affecting resilience among pharmacy students and faculty members and has received ethics approval from University College London, in the United Kingdom. At the time of writing, the study was in the data collection phase, using the Connor–Davidson Resilience Scale 25 (CD-RISC-25)©. It is anticipated that the study results will provide evidence-based recommendations to inform all stakeholders in pharmacy higher education institutions in the EMR of mechanisms and systems needed to enhance resilience and general-wellness of students and academics. Moreover, fostering resilience has been positively attributed to academic engagement and achievement and contributes to student mental health and well-being. Therefore, understanding the magnitude and associated factors of resilience amongst pharmacy students and faculty members in the EMR, would be helpful to university administrators and policymakers in the region.

### 3.5. Exploring Pharmacy Student and Pharmacist Resilience, Well-Being and Burnout in the United Kingdom

In the UK, GRiT steering group members from three universities (Reading, Manchester and Cardiff) are collaborating on research focusing on pharmacy student and pharmacist resilience, well-being and burnout. In 2019, the Academic resilience scale (ARS-30) and the Warwick–Edinburgh Mental Well-Being Scale (WEMWBS) were completed by all Master of Pharmacy (MPharm) students across the three institutions. A total of 1161 undergraduate pharmacy students completed the measures, with low resilience and well-being scores correlated. Within each university, these findings have supported implementation of student self-management and well-being programs.

In June 2020, a longitudinal cross-sectional online survey was distributed via social media to explore UK pharmacists’ response to the COVID-19 pandemic. The online survey consisted of four validated scales: Connor–Davidson Resilience Scale (CD-RISC 10), Oldenburg Burnout Inventory (OBI), Short Warwick–Edinburgh Mental Well-Being Scale (SWEMWB0) and the Recovery Experiences Questionnaire (REQ). In addition, open questions were asked regarding work challenges, positive changes to working practice, and most useful support at work, alongside participant demographic data. A total of 202 questionnaires were completed, with representation from all patient-facing sectors of pharmacy practice (primary and secondary care). Well-being, resilience, burnout and recovery scores were all found to be lower than population norms. Key themes identified included changes in working practice, emotional and physical responses, inter- and intra-professional working, organizational tensions, staffing and work–life balance. A second round of data collection was undertaken in October 2020, with analysis ongoing.

Establishing a smaller regional working group within GRiT has proven to be a successful and supportive collaboration of like-minded researchers, with progress and outputs reported back to GRiT within the monthly meetings.

### 3.6. Modifying Intern Exams to Reduce Stress and Facilitate Registration Progress in Australia

The pharmacy internship year is demanding, with requirements to complete a required number of practice hours, an Internship Training program as well as undertaking the registration examinations, which are in two parts—a written and an oral exam. COVID-19 placed additional significant stress on the 2020 cohort of interns due to disruptions to their supervised hours and their ability to undertake face-to-face examinations due to strict lockdown requirements in the first wave (Australia wide) and second wave (in Victoria).

The Australian Pharmacy Council (APC), the accreditation authority for pharmacy education and training, and the Pharmacy Board of Australia, the regulator for pharmacists, recognized the potential impact of COVID-19 on pharmacy interns, and worked together to modify practice hour requirements, timing, length and delivery mode of the examinations. The APC Intern Written Examination, previously a 3 h computer-based exam, was modified to a 2 h examination and was offered through either remote online proctored or test center examination and delivered later in the intern year. The oral examination was conducted by video conference. These actions reduced the regulatory burden on interns, gave them more time to prepare and were implemented with extensive consultation and communication to ensure all interns were informed of the options that would suit them best. Two-thirds of interns elected to sit the October 2020 written examination in test centers, and one-third by remote proctoring. There were no significant differences in the outcomes across these groups. A survey of all candidates’ post-exam showed that 40% of those who chose online delivery did so due to safety concerns or travel restrictions during COVID-19. The result of a collaborative approach by the accreditor and regulator resulted in no delay to the 2020 interns being able to register, thus reducing the stress and anxiety for both the interns and the broader pharmacy workforce.

### 3.7. Opportunities for Research Collaboration

Several GRiT members working within Australian Hospitals identified the need to investigate the resilience of employees during the COVID-19 pandemic. The Royal Brisbane and Women’s Hospital (Queensland Australia) together with Monash Health (Victoria Australia) pharmacy departments, in collaboration with psychologists and statisticians, developed a multisite study to investigate Pharmacy Workforce Resilience during the COVID-19 pandemic. The study received full ethics and governance approval at both sites and was seen as a priority area for research in the respective organizations. The study was a prospective longitudinal study involving survey and semi-structured interview techniques. Questions were developed that related to participant resilience, coping strategies, recovery experiences and burnout. The survey was designed to be administered to participants three times during the COVID-19 pandemic. The first survey was delivered in July 2020 and the final survey in December 2020. The results are currently being analyzed but, as a result of participant comments in the surveys and semi structured interviews, a commitment has been made in both pharmacy departments to develop resilience training and education material for the pharmacy workforce.

### 3.8. Collaboration to Seek Development Funding

The motivated academics, practitioners, and regulators who are involved in GRiT benefit considerably from the opportunity to network and share promising practices. However, they recognized that the group’s full potential is limited by lack of funding and other support necessary to undertake high-quality research and implement cutting-edge programming to support resilience across the profession. While there is general recognition of the importance of professional resilience in most jurisdictions, there is limited funding available for research and development work. Where funding is available it is most often targeted to other, larger professions such as medicine or nursing. The broad networks of GRiT members resulted in opportunities to meet with potential funders, primarily from the pharmaceutical industry. While GRiT is neither a formal organization nor a structured group, its culture is collectivist, inclusive, and collaborative in orientation. The group selected three senior members to represent GRiT at meetings with potential funders. To ensure collective objectives were clearly presented and adequately represented, a written funding proposal was developed and circulated to all GRiT members for feedback and commentary. A collective decision was made that funding requests should focus on creating a sustainable infrastructure for the group’s on-going viability and support resilience researchers and practitioners in their work directly related to pharmacy practice. The value of a global collaborative representing different sectors of the profession, with a culture of collective decision making and willingness to openly share findings and resources, was seen as a significant asset. A key learning from this process has been the limitations associated with the informal nature of GRiT. Funders must have certainty and accountability from recipients that funding will be used appropriately. Despite these barriers, GRiT has been successful in attracting funding to support the development of CPD material for the pharmacy profession in resilience training and education, and welcomes the opportunity to collaborate with any organizations interested in progressing the resilience agenda within pharmacy.

## 4. Discussion

GRiT has developed a community of practice by bringing together like-minded people, interested and passionate about raising awareness for resilience development and training within the pharmacy profession. It has facilitated the embedding of the concept of resilience into undergraduate and postgraduate curricula. GRiT is currently an informal group of individuals who participate voluntarily in their own time. We decided, soon after formation, that the group would function well as a community of practice. Communities of practice are groups of people who share a concern or a passion for something they do and learn how to do it better as they interact regularly. It is recognized that a community of practice generally has three characteristics: the domain, the community and the practice [15].

### 4.1. The Domain

GRiT is not simply a network of people, but rather has an identity defined by a shared domain of interest about resilience within the pharmacy profession. The group has a defined global membership that meet regularly via video conference. The group does not necessarily see themselves as experts in the field but rather are committed to developing knowledge and skills in the area.

### 4.2. The Community

GRiT meetings occur every month and have standing agenda items that allow individuals to provide updates on their work, and to seek support and advice for progressing their ideas. Information is freely shared, and the terms of reference for the group require that information is not shared outside the group unless permission is sought beforehand. GRiT members regularly learn from each other, motivate each other and collaborate when the opportunity arises.

### 4.3. The Practice

GRiT members are all practitioners coming from all aspects of the pharmacy profession. We have pharmacy undergraduate and postgraduate educators, practicing hospital and community pharmacists, pharmacist regulators and pharmacist leaders. We have developed and continue to gather resources, experiences, and tools that are openly shared by the group.

### 4.4. Challenges

GRiT has faced some challenges over their first 18 months. Organizing suitable and socially acceptable times to meet across nine time zones has been a challenge. This is further complicated by day light saving changes occurring in some countries at different times. It is accepted that not all members can join a meeting every month owing to the times zones. Equity has been established by altering times every other month to suit western and eastern hemispheres. However, to maintain some consistency, the meetings occur during the last week of every month. This approach has been successful and received positive feedback from the group. Another challenge faced by GRiT was that posed by submitting the necessary ethics applications to local Human Research Ethics Committees to undertake research studies. Different processes and procedures exist to obtain ethical approval in different countries, making it challenging for the group to work on a single project. It is an aspiration of the group to collaborate at some stage on a single project and the group continues to learn and problem solve, how to navigate the necessary ethics procedures to enable this to happen.

### 4.5. Future

GRiT continues to grow and develop, and future activities will focus on strengthening the community of practice. In addition to providing an intentional time and space for discussions on resilience, GRiT will continue to provide a medium for exploring resilience through various lenses. The shared commitment of the group allows for discourse to push boundaries and challenge the ways in which we explore the intersection of resilience with other factors, particularly when it comes to health equity and racial justice. As such, the group intends to explore resilience not just as a one-dimensional understanding, but one that accounts for the privilege and accountability, or lack thereof, in recognizing resilience as a multidimensional social process. The group proposes to strengthen strategic plans and coordination in order to harness the power of the collective. GRiT will continue to share knowledge and skills within the community of practice but also to the wider pharmacy profession using a variety of platforms. In addition, the group aims to continue to be innovative, creating new knowledge and ideas for resilience development and training. In the longer-term direction of the group will be developed to address strategic and emergent requirements in the area of developing resilience across pharmacy.

## 5. Conclusions

This community of practice demonstrates what can be achieved when like-minded practitioners come together with a common interest and passion. Despite the challenges faced by the global nature of the group, GRiT continues to grow and develop for the purpose of learning, support and collaboration. This enables research in resilience and the development of material to enhance resilience training and development at all levels of the pharmacy spectrum, from development in health professionals to developing and delivering evidence-based programs to undergraduate and postgraduate students.

## Figures and Tables

**Table 1 pharmacy-09-00110-t001:** Community of practice activities undertaken by GRiT.

Activities	Examples from GRiT
Coordination and strategy	Chaired monthly meetings, clear agendas with actions and timelines.
Problem solving	Open dialogue providing opportunities to share and discuss challenges facing group members.
Request for information	Drawing on members for information about work conducted on resilience, including ethics application and published articles.
Seeking experience	Drawing on members experience about tools used to evaluate and measure resilience.
Reusing assets	Sharing study protocols to allow members to adapt to their environment and country. Sharing educational materials for curriculum development.
Building an argument	Brainstorming ideas, discussing priority areas and writing funding proposals.
Growing confidence	Provision of support and motivation for members to conduct their own studies and education events relevant to their environment.
Discussing developments	Regular opportunities for members to provide updates on their work in the form of formal presentations or informal discussion at meeting.
Documenting and presenting projects	Publication of projects and involvement in international webinars.
Visits	Difficult during COVID-19 pandemic but future opportunities are planned.

## Data Availability

Not applicable.

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
