# Peer review of "Developing a Global Community of Practice for Pharmacy Workforce Resilience—Meet GRiT"

_pharmacy, 2021, doi:10.3390/pharmacy9020110_

Round 1
Reviewer 1 Report
Overall, this was a thorough review of the efforts of a global group on addressing the important issue of resilience. It is encouraging to see these sorts of initiatives within the profession of pharmacy. The global discussions on this topic make this article even more impressive. My comments are minor and intended only to clarify and strengthen the manuscript.
- In the Abstract, the term "community of practice" in the final sentence is not capitalized like it is earlier. Should it be here? I recognize there are discussions about communities of practice in general and then describing the GRiT community of practice.
- Please consider defining pre-registration as part of the inclusion criteria. As a practitioner in the US, I am not familiar with this terminology.
- Under Results, please consider restructuring the sentence that begins with "Lave and Wenger" to improve readability.
- Under "Faculty of Pharmacy and Pharmaceutical Sciences, Monash Univ," there are two examples that are provided. The last sentences for each of these examples say "the topic of resilience generated meaningful discussion between the students and between students and skills coaches" as well as "There was a lot of energy in the room and the feedback from the class was overwhelmingly positive." How were these findings determined? Is this anecdotal? If so, please include this language. If not, please include objective findings only OR remove the information from the manuscript.
- Under "Challenges," what does "obtaining ethics" mean? I am not familiar with this phrasing as it relates to ethics. Are you suggesting it was a challenge to align ethical considerations?
Reviewer 2 Report
This is, i believe are very timely article. I like the fact that the primary participants represent a global community of Pharmacy Professionals.
I would be interested in understanding or seeing a little more detail on the rational for the two types of membership. Also, wee there any attempts at reaching out to potential members in the United States? Why or why not (just wondering as I think this would be of interest to several professional organizations and Schools/Colleges of Pharmacy).
I think this is a very well thought out article and I definitely think a very worthwhile project to continue. I look forward to further areas of interest.
Reviewer 3 Report
This study describe the development of resilience within the pharmacy workforce by GRiT, the global nature of the group. Resilience of healthcare professionals is an important issue, especially in the case of COVID-19. This study is useful as an introduction to efforts for resilience development.
Minor point
Are there any reports of the communities of practice like Grit, including other healthcare professionals? If so, consider comparing them.
RESULT 3.6
The relationship between GRIT and APC is not described. Please explain this point.
